# Prediction of Anti-Cancer Drug-Induced Pneumonia in Lung Cancer Patients: Novel High-Resolution Computed Tomography Fibrosis Scoring

**DOI:** 10.3390/jcm9041045

**Published:** 2020-04-07

**Authors:** Hiroshi Gyotoku, Hiroyuki Yamaguchi, Hiroshi Ishimoto, Shuntaro Sato, Hirokazu Taniguchi, Hiroaki Senju, Tomoyuki Kakugawa, Katsumi Nakatomi, Noriho Sakamoto, Minoru Fukuda, Yasushi Obase, Hiroshi Soda, Kazuto Ashizawa, Hiroshi Mukae

**Affiliations:** 1Department of Respiratory Medicine, Nagasaki University Graduate School of Biomedical Sciences, Nagasaki University, Nagasaki 852-8523, Japanhirokazu_pc@hotmail.co.jp (H.T.);; 2Clinical Research Center, Nagasaki University Hospital, Nagasaki 852-8501, Japan; 3Department of Pulmonology and Gerontology, Yamaguchi University Graduate School of Medicine, Ube 755-0046, Japan; 4Clinical Oncology Center, Nagasaki University Hospital, Nagasaki 852-8501, Japan; 5Department of Respiratory Medicine, Sasebo City General Hospital, Nagasaki 857-8511, Japan

**Keywords:** lung cancer, interstitial lung disease, pulmonary fibrosis, radiology and other imaging

## Abstract

Background and objective: Pre-existing interstitial lung disease (ILD) in lung cancer patients is considered a risk factor for anti-cancer drug-induced pneumonia; however, a method for evaluating ILD, including mild cases, has not yet been established. We aimed to elucidate whether the quantitative high-resolution computed tomography fibrosis score (HFS) is correlated with the risk of anti-cancer drug-induced pneumonia in lung cancer patients, even in those with mild pre-existing ILD. Methods: The retrospective single-institute study cohort comprised 214 lung cancer patients who underwent chemotherapy between April 2013 and March 2016. The HFS quantitatively evaluated the grade of pre-existing ILD. We extracted data regarding age, sex, smoking history, and coexisting factors that could affect the incidence of anti-cancer drug-induced pneumonia. Cox proportional hazard models were used to analyze the effects of the HFS and other factors on the risk of anti-cancer drug-induced pneumonia. Results: Pre-existing ILD was detected in 61 (29%) of 214 patients, while honeycombing and traction bronchiectasis were observed in only 15 (7.0%) and 10 (4.7%) patients, respectively. Anti-cancer drug-induced pneumonia developed in 19 (8.9%) patients. The risk of anti-cancer drug-induced pneumonia increased in proportion to the HFS (hazard ratio, 1.16 per point; 95% confidence interval, 1.09–1.22; *p* < 0.0001). Conclusions: The quantitative HFS was correlated with the risk of developing anti-cancer drug-induced pneumonia in lung cancer patients, even in the absence of honeycombing or traction bronchiectasis. The quantitative HFS may lead to better management of lung cancer patients with pre-existing ILD.

## 1. Introduction

Anti-cancer drug-induced pneumonia is a potentially fatal disease in lung cancer patients. Pre-existing chronic interstitial lung disease (ILD) is considered a risk factor for anti-cancer drug-induced pneumonia, as is being male, being elderly, having a poor Eastern Cooperative Oncology Group performance status (ECOG-PS), a history of smoking, and low forced vital capacity [1,2,3,4]. The American Thoracic Society/European Respiratory Society/Japanese Respiratory Society/Latin American Thoracic Association guidelines for the classification of idiopathic interstitial pneumonia (IIP) are often employed for evaluating ILD [5,6,7]. Several clinical trials of anti-cancer drugs have included lung cancer patients with ILD, which was evaluated according to the IIP guidelines [8,9,10,11,12].

However, the IIP guidelines do not include and evaluate mild pre-existing ILD without honeycombing or traction bronchiectasis. The 2018 idiopathic pulmonary fibrosis (IPF) guidelines define mild ILD as “indeterminate for usual interstitial pneumonia (UIP) HRCT pattern” [13]. Nevertheless, few studies regarding anti-cancer drug-induced pneumonia have referred to mild pre-existing ILD [1,2]. Thus, a quantitative method for evaluating pre-existing ILD, including mild ILD, has not yet been established.

The quantitative high-resolution computed tomography (HRCT) scores of acute interstitial pneumonia and acute respiratory distress syndrome (ARDS) have been reported to correlate with pathology and prognosis [14,15,16,17]. The HRCT fibrosis score (HFS) was modified from this score in order to evaluate lung fibrosis easily, with an increased HFS over the course of 6 months indicating poor prognosis [18]. We hypothesized that the HFS could reflect the degree of the lung fibrosing process, resulting in the exact evaluation of the risk of anti-cancer drug-induced pneumonia in lung cancer patients with pre-existing ILD. This retrospective cohort study was designed to determine whether the quantitative HFS correlates with the risk of anti-cancer drug-induced pneumonia in lung cancer patients, including those with mild pre-existing ILD.

## 2. Methods

### 2.1. Study Design

This retrospective, single-institute, cohort study was performed at the Department of Respiratory Medicine, Nagasaki University Hospital. Patients were enrolled based on the following three inclusion criteria: admitted to our department between April 2013 and March 2016; pathologically diagnosed with lung cancer; and receiving anti-cancer drugs, including cytotoxic drugs and molecular-target agents. Patients were excluded from the analysis if they irregularly received adjuvant chemotherapy with oral tegafur–uracil alone or if >50% of the lung fields could not be evaluated primarily due to the extensive cancer lesions or postoperative status.

Eligible patients were extracted from lung cancer patients using the hospital’s electronic medical record system. Anti-cancer drugs were administered in various clinical conditions, including conventional or adjuvant chemotherapy, and concurrent or sequential radiotherapy. All patients were followed up with until December 2017. A systemic follow-up survey of the lesions was performed by physical examination, chest radiography, and blood tests at least once a month. Chest HRCT was routinely conducted as scheduled for outpatient follow-up. The censored cases were defined as death, transferring hospital, lost to follow-up, or the initiation of immune checkpoint inhibitors.

### 2.2. Chest HRCT Scoring System

HRCT (Aquilion ONE^TM^, Canon Medical Systems, Ohtawara, Japan) scans were obtained with 0.5 mm collimation and a 1 mm slice thickness at 1 mm intervals from the lung apices to the bases in the supine position at full inspiration. Two experienced pulmonologists (H.G. and H.I. with 11 and 19 years of experience, respectively), who were blinded to the clinical data, individually assessed the degree of pre-existing ILD using the HFS.

In detail, the HFS was calculated in three areas of each lung before the first administration of anti-cancer drugs: the level of the carina, the level of the right inferior pulmonary vein, and the middle of the two levels. First, the HRCT findings were scored as follows: normal attenuation (score 1), reticular abnormality (score 2), both reticular abnormality and traction bronchiectasis (score 3), and honeycombing (score 4). We did not evaluate extensive pure ground glass opacity, and we organized the lesions after pneumonia, operation, and radiotherapy. Second, the extent of the interstitial abnormalities was estimated as the percentage of the 5% intervals. We multiplied the score and the extent percentage and summed the points for each of the six areas. Finally, we averaged the summed points of the six areas. ILD was categorized as no ILD (HFS = 100), mild ILD (HFS = 101–200), moderate ILD (HFS = 201–300), and severe ILD (HFS = 301–400).

Furthermore, the two independent pulmonologists also evaluated the degree of emphysema using the Goddard score (GS) for the six areas analyzed above [19]. The extent of LAA (low-attenuation areas) was scored as follows: < 5% LAA (score of 0), 6%–25% LAA (score of 1), 26%–50% LAA (score of 2), 51%–75% (score of 3), and > 75% LAA (score of 4). The six scores were summed, and emphysema was graded as no emphysema (GS = 0), mild emphysema (GS = 1–7), moderate emphysema (GS = 8–15), and severe emphysema (GS = 16–24).

### 2.3. Examples of HFS and GS in Lung Cancer Patients with Pre-Existing ILD

Two representative patients were evaluated using the HFS (Figure 1). Regarding the HFS, Patient A scored 104 points, and patient B scored 122.5 points. The details are provided in Figure 1. Regarding the GS, patient A scored 0 points according to the two pulmonologists. Patient B was awarded 18 points by Dr. H.G. Each of six lung fields was evaluated as score 3; 3 × 6 = 18. In contrast, Patient B was awarded 15 points by Dr. H.I. Overall, patient B received an average GS of 16.5, indicating severe emphysema.

### 2.4. Anti-Cancer Drug-Induced Pneumonia

The definition of anti-cancer drug-induced pneumonia was interstitial lung disease developed from anti-cancer drug exposure to 60 days after the final anti-cancer treatment. We determined the diagnosis based on clinical and radiological findings and excluded apparent pulmonary infection, radiation-induced lung injury, and heart failure. The evaluation of anti-cancer drug-induced pneumonia was defined according to the Common Terminology Criteria for Adverse Events (CTCAE) version 4.0 from the National Cancer Institute (Bethesda, MD) [20]. The time to onset of anti-cancer drug-induced pneumonia was defined as the interval between the first administration of anti-cancer drugs and the diagnosis of anti-cancer drug-induced pneumonia.

### 2.5. Clinical Data Collection

We collected other clinical data, including age; sex; ECOG-PS; smoking history; tumor histology; clinical stage; epidermal growth factor receptor mutations; anaplastic lymphoma kinase rearrangements; and previous chemotherapy, thoracic surgery, and thoracic radiation.

### 2.6. Statistical Analysis

The inter-observer variations in the HFS and GS were assessed using intra-class correlation coefficients (ICC). The prevalence of pre-existing ILD and anti-cancer drug-induced pneumonia was accompanied by 95% confidence intervals (CIs), and the difference was evaluated by chi-square test. The cumulative incidence of anti-cancer drug-induced pneumonia, which was stratified by HFS of >110, 110 ≥ HFS > 100, and HFS = 100, was assessed via the Kaplan–Meier method and the trend log-rank test. We evaluated the hazard ratio (HR) and 95% CI of each potential risk factor for anti-cancer drug-induced pneumonia by simple Cox proportional hazard analysis. We selected variables according to a literature review and clinical expertise. We confirmed whether HFS (per point) may be affected by another potential risk factor using replaced multiple Cox proportional hazards analyses.

The plotting of Kaplan–Meier curves and the trend log-rank test were conducted using GraphPad Prism version 7 (GraphPad Software Inc., La Jolla, CA, USA). The other statistical analyses were performed using JMP pro version 14 (SAS Institute, Cary, NC, USA). A two-tailed *p* value of < 0.05 was considered statistically significant.

### 2.7. Ethical Considerations

This study was performed in accordance with the Declaration of Helsinki. The institutional review board of Nagasaki University Hospital approved this study protocol (approval number 17032713-3). Informed consent was obtained from the patients by an opt-out system on the hospital website, in accordance with the ethical guideline presented by the Ministry of Health, Labor, and Welfare in Japan. This trial was registered with the University Hospital Medical Information Network in Japan (UMIN), registry number UMIN000026964.

## 3. Results

### 3.1. Patient Characteristics and HFS

A total of 214 lung cancer patients were extracted and analyzed from 1280 patients with respiratory diseases (Figure 2). The study population was composed of approximately 70% males and ever smokers (Table 1). The inter-observer agreement on the HRCT findings was high for the HFS (ICC = 0.96) and GS (ICC = 0.97) between the two pulmonologists. The prevalence of pre-existing ILD was 28.5% (61/214 cases; 95% CI, 22.5–34.5) in lung cancer patients undergoing anti-cancer drug therapy. Incidentally, all pre-existing ILDs scored by the HFS were mild grade. Honeycombing and traction bronchiectasis were observed in only 15 (7.0%) and 10 (4.7%) of 214 cases, respectively.

The prevalence of anti-cancer drug-induced pneumonia was 8.9% (19/214 cases; 95% CI: 5.0–12.8). Anti-cancer drug-induced pneumonia was likely to develop in patients with pre-existing ILD (Figure 3). Although anti-cancer drug-induced pneumonia occurred twice in two patients with mild ILD, we analyzed the first event only. The prevalence of anti-cancer drug-induced pneumonia was 24.6% (15/61 cases; 95% CI, 13.9–35.3) in patients with pre-existing ILD, and 2.6% (4/153 cases; 95% CI, −0.4–5.6) in those without ILD. The development of anti-cancer drug-induced pneumonia was more frequent in patients with pre-existing ILD than in those without ILD (*p* < 0.0001).

### 3.2. Potential Risk Factors of Anti-Cancer Drug-Induced Pneumonia

The cumulative risk of anti-cancer drug-induced pneumonia was significantly increased in patients with a higher HFS (*p* < 0.0001, Figure 4). The simple Cox proportional hazard analysis showed a risk factor for anti-cancer drug-induced pneumonia (Table 2A); a high HFS and being male were significant risk factors for anti-cancer drug-induced pneumonia. The HFS in particular was a prominent risk factor; an increase of one HFS point was associated with a 16% increased risk of developing anti-cancer drug-induced pneumonia (HR, 1.16; 95% CI, 1.09–1.22; *p* < 0.001). Smoking history and high GS (indicating severe emphysema) were not associated with a risk of anti-cancer drug-induced pneumonia. The robustness of the HFS as a risk factor remained constant in the replaced multiple Cox proportional hazard analyses of other potential risk factors (Table 2B). A multivariate Cox proportional hazard analysis was not performed, because the simple and replaced multiple Cox proportional hazards analyses revealed that the HFS was a prominent risk factor of anti-cancer drug-induced pneumonia.

### 3.3. Characteristics of Patients with Anti-Cancer Drug-Induced Pneumonia

The characteristics of the 19 patients with anti-cancer drug-induced pneumonia are listed in Appendix A. Although previous radiotherapy did not confer a risk for anti-cancer drug-induced pneumonia (Table 2A), 10 of 19 these patients had received radiotherapy. The development of radiological interstitial abnormalities in the ten patients did not correspond to the exposure to radiation. The HFS did not correlate with the grade of anti-cancer drug-induced pneumonia. Specific regimens were not associated with the risk of anti-cancer drug-induced pneumonia.

## 4. Discussion

To the best of our knowledge, the present study is the first to provide the following important findings. The quantitative HFS is positively correlated with the risk of anti-cancer drug-induced pneumonia in lung cancer patients. In particular, the HFS was associated with a risk of anti-cancer drug-induced pneumonia in patients with mild pre-existing ILD, even in the absence of honeycombing or traction bronchiectasis.

The current IIP guidelines do not assess the severity of pre-existing ILD, particularly mild ILD. IPF diagnosed using the IIP guidelines was observed in 2%–8% of lung cancer patients [21,22,23]. In contrast, 28.5% of lung cancer patients that were administered anti-cancer drugs had mild pre-existing ILD confirmed by the HFS in the present study. In two computed tomography lung cancer screening trials that defined the patterns of mild interstitial lung abnormalities, ILD was detected in 15.7%–21.2% of patients with a history of smoking [24,25]. Since the IIP guidelines do not necessarily consider the radiological process of pulmonary fibrosis, mild ILD may have been overlooked. Precise evaluation of the risk of anti-cancer drug-induced pneumonia requires a quantitative method capable of detecting all levels of ILD, including mild ILD.

In the present study, the quantitative HFS was significantly correlated with the risk of anti-cancer drug-induced pneumonia. Little is known of the relationship between anti-cancer drugs and mild pre-existing ILD [1,2]. A nested case-control study investigated the risk factors for pre-existing ILD associated with chemotherapy and gefitinib [1]. Although this study referred to the severity of ILD, including mild grade, the severity was not clearly defined. The adjusted odds ratios indicated that patients with mild and moderate–severe ILD had an approximately five-fold increased risk of developing acute ILD compared to patients without ILD, suggesting no difference between mild and moderate–severe ILD. Another prospective cohort study identified the risk factors for pre-existing ILD associated with erlotinib [2]. This study evaluated only the extent of interstitial abnormalities, but not the fibrous pattern on HRCT. The study population of patients with pre-existing ILD predominantly comprised patients with mild ILD, in which abnormalities involved <5% of the bilateral lower lobes. The multivariate logistic regression analysis yielded an odds ratio of 4.0 (95% CI, 1.3–12.1) between the presence and absence of ILD. Thus, the present study first revealed that an increase of HFS point was associated with a 16% increased risk of anti-cancer drug-induced pneumonia, even in mild ILD without honeycombing and traction bronchiectasis.

Pre-existing mild ILD might induce drug-induced pneumonia because a histological UIP pattern may be involved. The 2018 IPF guidelines categorize mild ILD as an indeterminate for the UIP HRCT pattern (early UIP pattern) [13]. This category includes the subset of patients with reticular patterns in the absence of honeycombing and traction bronchiectasis, predominantly in the subpleural and basal fields. Lung cancer patients with UIP HRCT patterns exhibited exacerbation of ILD more frequently than did those with non-UIP HRCT patterns [3]. In precision medicine, many novel anti-cancer drugs have been developed, some of which are likely to cause drug-induced pneumonia. The early recognition of the subtle fibrosing findings on HRCT may enable physicians to provide better management of lung cancer patients receiving anti-cancer drugs.

The present study had several limitations. First, all cases of pre-existing ILD incidentally included mild cases. Thus, we did not necessarily verify the utility of the HFS for moderate and severe ILD. However, a previous case-control study revealed that the odds ratio of the development of anti-cancer drug-induced pneumonia was high in patients with moderate/severe pre-existing ILD as well as those with mild ILD [1]. Second, the pulmonary function test results could not be evaluated, due to missing values in clinical practice. Another quantitative HRCT scoring system has been replaced by the diffusion capacity of carbon monoxide test for the prognosis with IPF patients [26]. Third, this was a single institutional analysis, and the number of anti-cancer drug-induced pneumonia cases was small. The present findings are therefore potentially subject to selection bias. Further prospective investigations involving large cohorts are required to confirm our findings.

In conclusion, HFS may correlate with the risk of anti-cancer drug-induced pneumonia in lung cancer patients, even in the absence of honeycombing or traction bronchiectasis. This quantitative HFS could be a predictor of anti-cancer drug-induced pneumonia, which could improve the management of lung cancer patients with ILD.

## Figures and Tables

**Figure 1 jcm-09-01045-f001:**
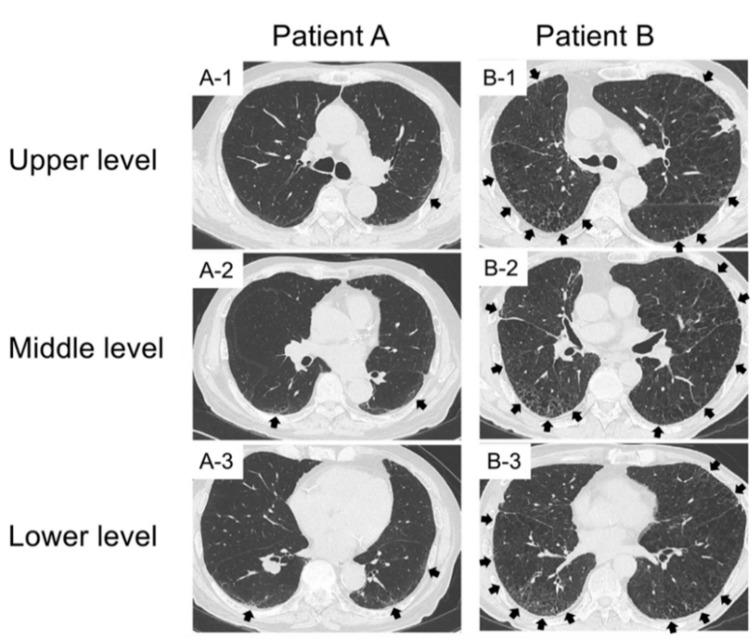
Imaging findings in two representative patients evaluated using the high-resolution computed tomography fibrosis score. The black arrows indicate reticular abnormality (score 2). Patient A scored 104 points, and patient B scored 122.5 points. The scoring of each patient was as follows: Patient A scored 105 points, as evaluated by Dr. H.G. In detail, the (**A-1**) right field was evaluated as 100% of score 1; 100 × 1 = 100. The A-1 left, (**A-2**) right, (**A-2**) left, and (**A-3**) right fields were evaluated as 95% of score 1 and 5% of score 2; (95 × 1) + (5 × 2) = 105. The (**A-3**) left field was evaluated as 90% of score 1 and 10% of score 2; (90 × 1) + (10 × 2) = 110. Thus, the HFS was (100 × 1 + 105 × 4 + 110 × 1)/6 = 105. In contrast, patient A scored 103 points, as evaluated by Dr. H.I. Overall, patient A received an average high-resolution computed tomography fibrosis score (HFS) of 104. Patient B scored 125 points, as evaluated by Dr. H.G. In detail, the (**B-1**) left were evaluated as 80% of score 1 and 20% of score 2; (80 × 1) + (20 × 2) = 120. The (**B-1**) right, (**B-2**) right and left, and (**B-3**) left were evaluated as 75% of score 1 and 25% of score 2; (75 × 1) + (25 × 2) = 125. The (**B-3**) right was evaluated as 70% of score 1 and 30% of score 2; (70 × 1) + (30 × 2) = 130. Thus, the HFS was (120 × 1 + 125 × 4 + 130 × 1)/6 = 125. In contrast, patient B scored 120 points, as evaluated by Dr. H.I. Overall, patient A received an average HFS of 122.5.

**Figure 2 jcm-09-01045-f002:**
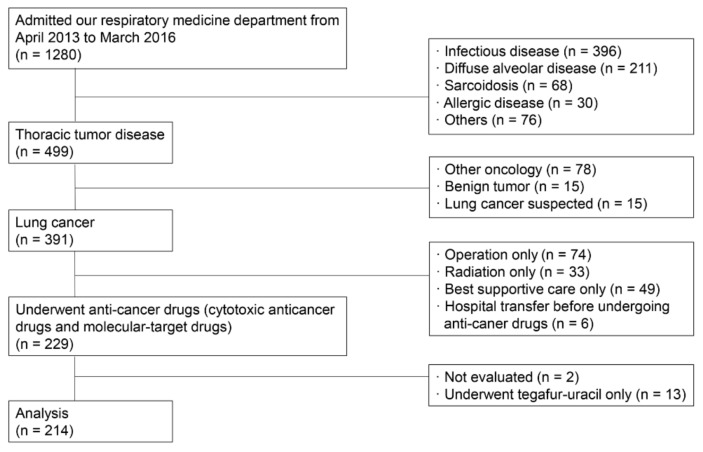
Study flow diagram.

**Figure 3 jcm-09-01045-f003:**
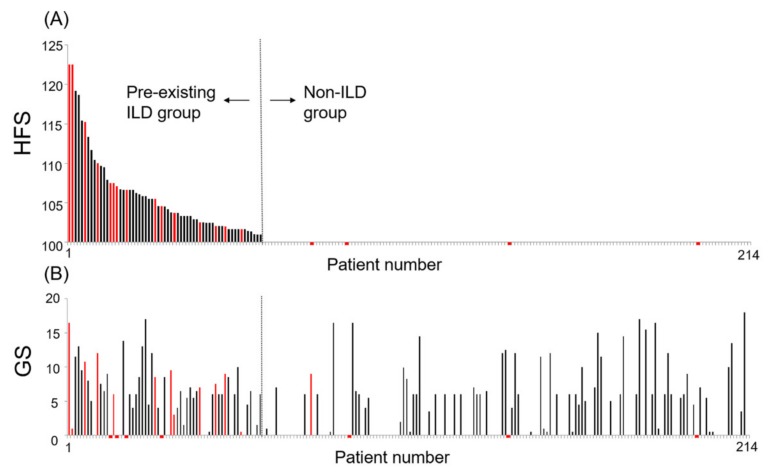
Bar graph of the incidence of anti-cancer drug induced pneumonia according to the high-resolution computed tomography fibrosis score and Goddard score. (**A**) Relationship between the high-resolution computed tomography fibrosis score (HFS) and the development of anti-cancer drug-induced pneumonia in 214 patients with lung cancer. The red bars represent the patients with anti-cancer drug-induced pneumonia, and the black bars represent those without. An HFS of >100 indicates the presence of pre-existing interstitial lung disease (ILD). (**B**) Relationship between the Goddard score (GS) and the development of anti-cancer drug-induced pneumonia in lung cancer patients. The red and black bars are defined as described above. Each pair of upper and lower panels represents the same patient.

**Figure 4 jcm-09-01045-f004:**
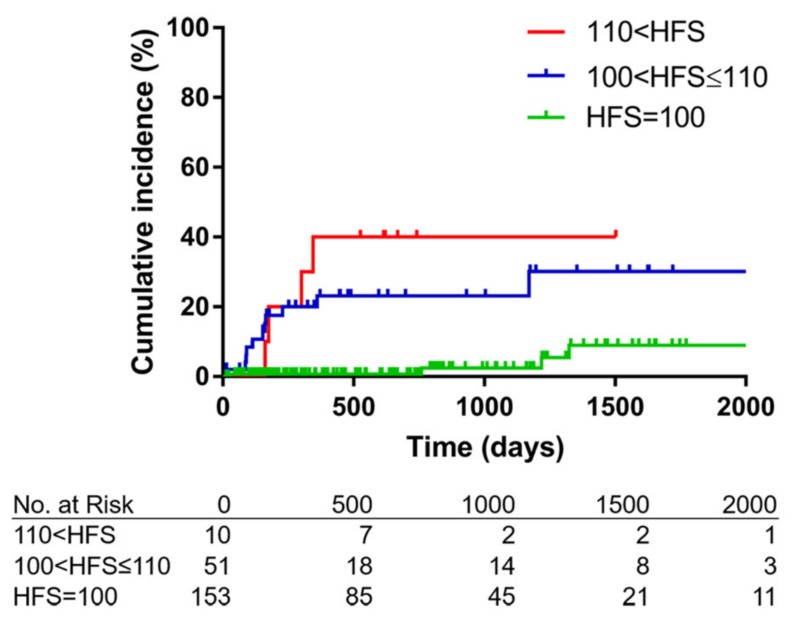
Kaplan–Meier estimated curves of the incidence of anti-cancer drug-induced pneumonia stratified by the high-resolution computed tomography fibrosis score. The Kaplan–Meier estimated curves indicated that the incidence of anti-cancer drug-induced pneumonia is increased in patients with higher high-resolution computed tomography fibrosis scores (HFSs) (*p* < 0.0001 according to the log-rank test).

**Table 1 jcm-09-01045-t001:** Characteristics of lung cancer patients who underwent anti-cancer drugs.

	Total (*n* = 214)
Age, range	67 (25–85)
Gender	
Male	144 (67%)
Female	70 (33%)
Smoking history	
Ever-smoker	148 (69%)
Never-smoker	66 (31%)
ECOG-PS	
0–1	209 (98%)
2–4	5 (2%)
Histology	
Adenocarcinoma	136 (64%)
Squamous cell carcinoma	35 (16%)
Small-cell lung cancer	31 (14%)
Others	12 (6%)
Clinical stage	
I	4 (2%)
II	14 (7%)
III	53 (25%)
IV, Recurrence	143 (67%)
Genetic abnormalities	
Wild type	151 (71%)
EGFR mutations	56 (26%)
ALK rearrangements	7 (3%)
GS	
Normal	93 (43%)
Mild	77 (36%)
Moderate	36 (17%)
Severe	8 (4%)

ECOG-PS, Eastern Cooperative Oncology Group performance status; EGFR, epidermal growth factor receptor; ALK, anaplastic lymphoma kinase; GS, Goddard score.

**Table jcm-09-01045-t002a:** (**A**)

	HR	95% CI	*p*-value
Age			
< 70	1.00		
≥ 70	1.37	0.51–3.42	0.51
Gender			
Female	1.00		
Male	3.29	1.09-14.18	0.034
ECOG-PS			
0–1	1.00		
2–4	3.61	0.20–17.67	0.30
Histology			
Adenocarcinoma	1.00		
Squamous-cell carcinoma	0.70	0.11–2.53	0.62
Small-cell carcinoma	0.72	0.11–2.62	0.65
Others	1.03	0.06–5.22	0.97
Clinical Stage			
I–II	n/c		
III	1.00		
IV, Recurrence	0.91	0.36–2.60	0.86
Smoking history			
No	1		
Yes	2.97	0.99-12.80	0.053
Genetic abnormalities ^†^			
No	1		
Yes	0.65	0.21–1.73	0.40
GS (per point)	1.06	0.97–1.15	0.20
HFS (per point)	1.16	1.09–1.22	< 0.0001
Operation			
No	1		
Yes	0.49	0.16–1.30	0.16
Radiation			
No	1		
Yes	1.66	0.66–4.14	0.28

^†^ Gene abnormalities includes anaplastic lymphoma kinase (ALK) and epidermal growth factor receptor (EGFR). HR, hazard ratio; CI, confidence interval; ECOG-PS, Eastern Cooperative Oncology Group performance status; n/c, not calculated; GS, Goddard score; HFS, HRCT fibrosis score.

**Table jcm-09-01045-t002b:** (**B**)

	HFS		
Variables	HR	95% CI	*p*
Simple			
HFS	1.16	1.09–1.22	< 0.0001

Replaced multiple			
Variable to adjust the effect of HFS			
Age	1.15	1.08–1.22	<0.0001
Gender	1.14	1.07–1.21	0.0002
ECOG-PS	1.16	1.09–1.22	<0.0001
Histology	1.16	1.09–1.23	<0.0001
Clinical stage	1.15	1.08–1.22	<0.0001
Smoking history	1.15	1.08–1.22	0.0001
Gene abnormalities	1.16	1.08–1.23	<0.0001
GS	1.16	1.08–1.23	<0.0001
Operation	1.16	1.09–1.22	<0.0001
Radiation	1.16	1.09–1.23	<0.0001

HFS, HRCT fibrosis score; HR, hazard ratio; CI, confidence interval; ECOG-PS, Eastern Cooperative Oncology Group performance status; GS, Goddard score.

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
