# Peer review of "Prediction of Anti-Cancer Drug-Induced Pneumonia in Lung Cancer Patients: Novel High-Resolution Computed Tomography Fibrosis Scoring"

_jcm, 2020, doi:10.3390/jcm9041045_

Round 1

Reviewer 1 Report

I thank the authors for this interesting and clinically important work.

Comments as follows:

Introduction

Page 2, line 44 – delete word “been”

Methods

Page 4, line 125 – please provide reference for Common Terminology Criteria for Adverse Events. Consider an in-text description of the definition of drug-related pneumonia.

Results

Page 4, line 155 – correct number of patients with respiratory diseases to 1280

Figure 4 is not referenced / discussed in the text – consider commenting on the initial steepness of the curve in patients with HFS over 110, compared to the more gradual increase over time in patients with HFS <110

Other comments

Are the authors surprised by lack of association between drug-induced pneumonia and radiation? Consider comment on this aspect.

The authors comment about an early UIP pattern being a possible explanation for the HRCT findings. How can you attribute lung changes to the drug and not natural progression of the ILD?

Author Response

Thank you very much for reviewing our manuscript. I am very pleased to see the favorable comments.

Introduction

  1. Page 2, line 44 – delete word “been”

Response: We have deleted the word “been” (Page 2, line 44) according to the comments from the reviewer.

Methods

  1. Page 4, line 125 – please provide reference for Common Terminology Criteria for Adverse Events. Consider an in-text description of the definition of drug-related pneumonia.

Response: We have added the reference (Page 4, line 125) as No.20 and we have rewritten the definition and explanations of anti-cancer drug pneumonia. We hope that the edited section clarifies a process of the diagnosis.

Results

  1. Page 4, line 155 – correct number of patients with respiratory diseases to 1280

Response: We have revised the number of patients to “1280” (Page4, line 155).

  1. Figure 4 is not referenced / discussed in the text – consider commenting on the initial steepness of the curve in patients with HFS over 110, compared to the more gradual increase over time in patients with HFS <110

Response: Thank you for a new viewpoint. Kenmotsu, et al. (reference No.3) described that UIP pattern were significant risk factors for cytotoxic chemotherapy-related exacerbation of interstitial lung disease. We guessed that high HFS score group included a more proportion of UIP. The patients with pre-existing UIP maybe tend to onset earlier. We should investigate further study about this point.

Other comments

  1. Are the authors surprised by lack of association between drug-induced pneumonia and radiation? Consider comment on this aspect.

Response: We were surprised as with the reviewer’s comment. We think that there are several reasons and biases because of retrospective study. First, clinicians often avoid chemoradiotherapy / radiotherapy for the patients with apparent interstitial lung shadows in clinical setting. Second, there might be included patients with low dose radiation exposure without full dose radiation consideration with poor PS, old age, and a difference of object.

  1. The authors comment about an early UIP pattern being a possible explanation for the HRCT findings. How can you attribute lung changes to the drug and not natural progression of the ILD?

Response: You have raised an important point. We agree that it is difficult to distinguish “drug induced” and “natural progression”. However, we limited the onset time range of anti-cancer drug induced pneumonia between initiation of first chemotherapy to 60 days after the final anti-cancer treatment in this study. Thus, we believed that the majority of patients diagnosed drug-induced pneumonia in this study were “anti-cancer drug- induced” patients. Furthermore, chemotherapy was avoided in progressive UIP case in general practice. Mild interstitial pneumonia case in this study had low probability of natural progression.

Reviewer 2 Report

Line 129. Did you collect information about the oral medication patients received before initiation and during administration of chemotherapy or target agents?
Line 136-137. Please check the wording of HFS >100...
Line 155. Please, confirm that you refer to 1280 patients
FIGURE 2: Correct "Anticancer drugs"
LINE 189. Do you consider the value of HR 1.2, although statistically significant, to be clinically relevant?

If the authors introduce the use of this score into clinical practice, prospectively, what approach will they take in managing patients at increased risk of pneumonitis? Drug selection by avoiding the most dangerous drugs? Close monitoring? Other measures?

To what extent are these results of interest for the management of patients treated with immunotherapy, which is currently used in a high percentage of patients with lung cancer?

Author Response

Reply to the Comments of Reviewer 2:

We appreciate your reviewing our manuscript and offering valuable comments. We agree with the reviewer’s comments and have revised the original manuscript.

  1. Line 129. Did you collect information about the oral medication patients received before initiation and during administration of chemotherapy or target agents?

Response: We did not collect the oral medication information without oral steroid administration. No patients received continuous oral steroids in this study. As the reviewer’s comment, the possibility of drug induced pneumonia caused by other oral medication other than anti-cancer drugs did not completely excluded. However, we reviewed the pneumonia patients again, there were no cases that oral medication was doubted as suspected drug than anti-cancer drugs in this study.

  1. Line 136-137. Please check the wording of HFS >100...

Response: We have revised the text “>110, 110 ≥ HFS > 100, and HFS = 100” according to the reviewer’s comment.

  1. Line 155. Please, confirm that you refer to 1280 patients

Response: We have corrected the number of patients to 1280 (Page 4, line 165).

FIGURE 2: Correct "Anticancer drugs"

Response: Thank you for your suggestion. We believe that the term “anti-cancer drug” would be more appropriate. We would like to keep in accordance with the main text. Should I change it?

  1. LINE 189. Do you consider the value of HR 1.2, although statistically significant, to be clinically relevant?

Response: We have intended that one HFS point other than three HFS categorized groups was associated with a 16% increased risk. Thus, we think that it is clinically important. We have rewritten to be clarified our intension and added the word “(per point)” in the text (Page 4, line 149).

  1. If the authors introduce the use of this score into clinical practice, prospectively, what approach will they take in managing patients at increased risk of pneumonitis? Drug selection by avoiding the most dangerous drugs? Close monitoring? Other measures?

Response: We agree with you and recommend individual management according to the HFS score to the patients. For example, patients with high HFS score should avoid high risk drugs, and we recommend early and close monitoring of chest X-ray, CT, blood monitoring (KL-6, SP-D, etc.), and SpO2.

  1. To what extent are these results of interest for the management of patients treated with immunotherapy, which is currently used in a high percentage of patients with lung cancer?

Response: This is an important point. One retrospective analysis indicates that pre-existing ILD is a risk factor of anti-PD-1 related pneumonitis (Lung Cancer. 2018; 125: 212-217). In the study, the authors categorize ILD to six severity resembled to HFS scoring system, HFS could be a predictor of PD-1 related pneumonitis. We should investigate further study in order to confirm.